# Oral Iron for IBD Patients: Lessons Learned at Time of COVID-19 Pandemic

**DOI:** 10.3390/jcm9051536

**Published:** 2020-05-19

**Authors:** Ferdinando D’Amico, Laurent Peyrin-Biroulet, Silvio Danese

**Affiliations:** 1Department of Biomedical Sciences, Humanitas University, Pieve Emanuele, 20090 Milan, Italy; damico_ferdinando@libero.it; 2Department of Gastroenterology and Inserm NGERE U1256, University Hospital of Nancy, University of Lorraine, 54500 Vandoeuvre-lès-Nancy, France; peyrinbiroulet@gmail.com; 3IBD Center, Department of Gastroenterology, Humanitas Clinical and Research Center, IRCCS, Rozzano, 20089 Milan, Italy

**Keywords:** anemia, iron deficiency, oral iron, inflammatory bowel disease, COVID-19

## Abstract

Anemia is a frequent manifestation in patients with chronic inflammatory bowel disease (IBD) and requires tight monitoring and adequate supplementary therapy. Intravenous iron is the first-line treatment in subjects with moderate–severe anemia, active disease, or oral iron intolerance. On the other hand, oral iron is recommended in patients with mild anemia and inactive disease. However, during the current coronavirus pandemic, hospital activities have significantly changed, and all non-essential procedures, including non-urgent iron infusions, have been rescheduled. Oral iron, including both the traditional formulations with ferrous iron and the new ferric iron complexes, could constitute a valid alternative for anemia treatment. For this reason, we conducted a literature review, to summarize the scientific evidence on oral iron therapy in IBD patients with anemia.

## 1. Introduction

Crohn’s disease (CD) and ulcerative colitis (UC) are inflammatory bowel diseases (IBD) with a remitting and relapsing course [1,2]. IBD mainly causes gastrointestinal symptoms, but up to 50% of patients experience extraintestinal manifestations (EIMs) during their lifetime [3]. Anemia is the most frequent EIM in IBD subjects, with up to 70% of inpatients and 20% of outpatients being detected [4]. The World Health Organization (WHO)’s definition of anemia is also appropriate in the context of IBD (hemoglobin (Hb) levels <12 g/dL in non-pregnant women and <13 g/dL in men) [5,6]. Its pathogenesis is multifactorial: iron deficiency caused by malabsorption, blood loss, vitamin B12/folate deficiency in post-surgical patients, or the underlying chronic disease [5,7]. It negatively impacts the patient′s quality of life and his or her prognosis, and it is associated with an increased risk of hospitalization and surgery [8,9]. For this reason, in case of martial deficiency, iron should be supplemented, in order to normalize iron stores and hemoglobin levels [5]. There are two different iron formulations, the intravenous and the oral one. Intravenous iron is recommended as first-line treatment in IBD patients with hemoglobin levels <10 g/dL, signs of clinically active disease activity, or oral iron intolerance, while oral iron is indicated in patients with mild anemia and inactive disease [5]. Despite these recommendations, several studies [10,11] showed that oral iron was still the most frequent supplementation modality for the treatment of martial deficiency, and some controversies persist about the best route of iron administration [12]. Interestingly, a systematic review and metanalysis found no significant difference in correcting martial deficiency anemia between intravenous and oral iron in patients with IBD [12]. To date, no study has provided clear globally accepted evidence on this topic. The purpose of this review was to summarize the most recent and relevant data on oral iron therapy in the setting of iron-deficiency anemia of IBD patients.

## 2. Experimental Section

We searched on PubMed, EMBASE, Scopus, and Web of Science databases up to April 2020, in order to identify all studies assessing oral iron therapy in IBD patients. The following MESH terms alone or combined with the Boolean operators “AND” or “OR” were used: “anemia”, “martial deficiency”, “iron”, “oral”, “ferrous iron”, “ferric iron”, “ferric maltol”, “sucrosomial iron”, “treatment”, “therapy”, “supplementation”, “CD”, “Crohn’s disease”, “ulcerative colitis”, “UC”, “inflammatory bowel disease”, and “IBD”. No temporal or language restrictions were applied. We focused on full-text articles, even if abstracts were evaluated when relevant. Additional studies were identified through the careful investigation of the reference lists of the articles. Finally, the studies were included based on their relevance, after approval by all authors.

## 3. Results

### 3.1. Ferrous Oral Iron

The main characteristics of ferrous oral iron studies for the treatment of anemia in IBD patients are summarized in Table 1. Oral iron formulations mainly contain the ferrous form of iron (Fe^++^), as it has a greater bioavailability than the ferric variant [13]. The most used formulations are ferrous sulphate, ferrous gluconate, and ferrous fumarate [13]. The optimal oral iron dose is not known. Since a small percentage of oral iron is absorbed (10–25%), the recommended daily dose should not exceed 200 mg per day, to avoid the onset of adverse events [13]. Several studies addressed the treatment of anemia by comparing the different administration routes. A meta-analysis by Lee et al. analyzed data from 333 IBD patients recruited in three randomized clinical trials, assessing efficacy and safety of oral and intravenous iron formulations [14]. The primary outcome was the mean difference in Hb at the end of study compared to baseline. A greater improvement in Hb levels was achieved after infusion therapy, compared to oral administration, with a mean difference of 6.8 g/L (CI 0.9–12.7, *p* = 0.02) [14]. A higher increase in serum ferritin levels was also found in the intravenous group, compared to the oral one, with a mean difference of 109.7 μg/L (CI 5.37–214, *p* = 0.04). Adverse events leading to therapy discontinuation mostly occurred in patients treated with oral iron (odds ratio 6.2, CI 2.21–17.1), while no difference between the two groups was detected in terms of quality of life [14]. The systematic review by Nielsen et al. investigated 13 prospective studies with a follow-up of at least one month [15]. In patients with mild anemia (hemoglobin (Hb) > 10g / dL), no difference in efficacy (defined as normalization of Hb values) was detected between oral and intravenous iron [15]. Intravenous iron allowed for the attainment of a higher concentration of ferritin, but no difference in Hb values was revealed [15]. On the other hand, in subjects with more severe anemia (Hb < 10 g/dL) intravenous iron led to greater improvements in Hb values [15]. No significant correlation between oral/intravenous therapy and disease activity was found [15]. A meta-analysis by Abhyankar and Moss evaluated data from five randomized clinical trials, including 695 IBD patients [12]. The primary endpoint was the response to treatment, defined as an increase in Hb of >2 g/dL, while the secondary endpoints were mean change in Hb, mean change in serum ferritin, and withdrawal due to adverse events [12]. Oral and intravenous iron did not differ in terms of improving Hb values [12]. However, intravenous iron was associated with a significant increase in serum ferritin, compared to oral iron, with a mean difference of 88.7, in favor of the intravenous formulation (95% CI, 29–148; *p* = 0.003). In addition, fewer patients treated with intravenous iron discontinued therapy due to adverse events than the oral group (risk ratio: 0.4; 95% CI, 0.1–1.0; *p* = 0.05) [12]. Importantly, Bonovas et al. detected serious data extraction errors in the meta-analysis of Abhyankar and Moss regarding the total number of participants and the numbers of events, compromising the study results [4]. For this reason, they performed a new meta-analysis, using the same studies included by colleagues, and concluded that intravenous iron was more effective than oral ferrous iron in achieving an increase in Hb greater than 2 g/dL in IBD patients (odds ratio (OR): 1.57, 95% confidence interval (CI): 1.13–2.18) [4]. Furthermore, fewer gastrointestinal adverse events occurred in the intravenous group than in the oral group, leading to a lower discontinuation rate (OR: 0.27, 95% CI: 0.13–0.59) [4]. However, a higher rate of serious adverse events was reported among infusion patients, compared to those orally treated (OR: 4.57, 95% CI: 1.11–18.8) [4]. A prospective randomized trial (included in the previous meta-analyses) investigated the non-inferiority of intravenous iron isomaltoside 1000, compared with oral iron sulfate, for the treatment of iron deficiency anemia in IBD patients [16]. The study failed to demonstrate the non-inferiority of the intravenous formulation, and a not-significant trend of greater efficacy was found in orally supplemented patients [16]. Regarding the safety profile, both drugs were well tolerated, and no difference was recorded in terms of adverse events [16]. On the other hand, a meta-analysis of randomized controlled trials evaluated the safety profile of oral and intravenous iron [17]. A significantly higher incidence of gastrointestinal adverse events was detected in IBD patients treated with oral ferrous iron compared to those treated with intravenous formulation (OR = 3.14, 95% CI 1.34–7.36, *p* = 0.008, I2 = 0%) [17]. Patients’ views on iron supplementation were evaluated in 2014, by a European Federation of Crohn’s and Ulcerative Colitis Associations (EFCCA) survey [18]. Among the 631 respondents, a greater percentage of patients was treated with oral iron compared to intravenous iron (42% vs. 27%) [18]. Most patients in the oral group (74%) were not satisfied with the treatment, due to poor tolerability (66%) and loss of efficacy (51%), while three-quarters of the subjects undergoing infusion therapy were satisfied, well tolerated the drug, and believed it to be effective [18]. Another questionnaire distributed to 87 adult IBD outpatients reported the presence of adverse events in about 50% of patients, and in a third of cases, anemic subjects were unable to complete oral iron therapy [19]. Overall, both oral and intravenous iron are effective for the treatment of martial deficiency anemia, but the greater percentage of adverse events in orally treated patients invalidates patient tolerance and adherence to therapy.

### 3.2. Ferric Oral Iron

The main characteristics of ferric oral iron studies for the treatment of anemia in IBD patients are reported in Table 2.

### 3.3. Ferric Maltol

In recent years, a new oral iron therapy composed of a complex of ferric iron (Fe^+++^) and maltol, a sugar derivative, has been developed and is now available [29]. Free iron can damage the intestinal mucosa and modify the gut microbiota [29]. The ferric maltol complex makes iron stable and prevents the formation of iron hydroxide polymers, increasing iron bioavailability and reducing the risk of mucosal toxicity [29]. A randomized Phase 3 study evaluated the efficacy and safety of ferric maltol in IBD patients with iron-deficiency anemia who were unresponsive or intolerant to oral ferrous forms [20]. The drug allowed a rapid improvement in Hb values, since there was a mean Hb increase of 1.04 g/dL, compared to the placebo, after four weeks of treatment [20]. Furthermore, the mean Hb value in patients treated with ferric maltol was significantly higher than in the placebo group, after 12 weeks, with a mean Hb increase of 2.2 g/dL [20]. There was no difference in treatment compliance between the two groups (98% for both arms), and the percentage of adverse events between the experimental group and the control group was comparable (58% vs. 72%, respectively), indicating a good drug safety profile [20]. Long-term extension data of this Phase 3 trial confirmed the efficacy of ferric maltol [21]. In fact, 97 of the 111 patients who completed the first 12 study weeks were enrolled in an open-label trial, receiving ferric maltol 30 mg twice a day, for an additional 52 weeks [21]. Patients who were treated with the placebo in the previous phase were switched to oral iron [21]. A mean increase of 3.07 g/dL in Hb concentration was found after 64 weeks of treatment [21]. Interestingly, an Hb mean increase of 2.19 g/dL was reported in the switch population [21]. Resolution of anemia occurred in over 80% of patients, and adherence to therapy was maintained in 84% of cases [21]. The drug appeared safe, as only 27 patients (24%) experienced drug-related adverse events (abdominal pain, constipation, flatulence, and diarrhea), and most of them were mild or moderate in intensity [21]. Similarly, a recent Cochrane review highlighted an acceptable safety profile for ferric maltol, as fewer adverse events and serious adverse events were reported in patients treated with ferric maltol, as compared to the placebo (51% vs. 71% and 8% vs. 13%; low certainty evidence for both findings) [22]. Furthermore, preliminary data from an ongoing observational cohort study showed that two-thirds of the patients treated with ferric maltol achieved normalization of Hb levels after three months of oral supplementation in a real-life setting [23]. A real-life cohort study investigated IBD patient′s tolerability toward ferric maltol [24]. It was well tolerated by 67% of patients, and 50% of patients, who did not tolerate other oral forms of iron, tolerated ferric maltol [24]. Recently, an open-label, Phase 3B non-inferiority trial compared ferric maltol and intravenous iron, focusing on the loss of productivity in IBD patients [25]. Oral iron was not associated with a reduction in productivity [25]. In contrast, 50% of the intravenous group patients lost at least one day of work, and 6.7% of them lost four-to-six working days, causing up to 775 € of daily losses per patient [25]. Interestingly, the physical component summary of the short form health survey (SF-36) and the mental component summary (MCS) values were slightly higher (*p*-value not significant) in patients treated with oral medication, suggesting a greater improvement in quality of life at the end of the study period [26]. In addition, total treatment costs per patient were lower in the oral drug group than in the intravenous arm (302.27 € vs. 489.37 €, respectively) [27]. Influencing factors on the high cost of intravenous treatment were not only the greater cost of the drug, but also the number of hospitalizations/outpatient visits for each patient (2.30) [27].

### 3.4. Sucrosomial Iron

Sucrosomial iron is another formulation of oral iron which uses liposomal vesicles to transport the ferric iron [28]. It has been associated with an improved iron bioavailability and a lower drug dose than ferrous variants [28]. It consists of a ferric pyrophosphate core, a phospholipid bilayer membrane, and a sucrester matrix, which has a gastroprotective function [28]. In a pilot interventional study on 30 IBD patients, an improvement in Hb values was achieved by 86% of patients treated with 30 mg/day of sucrosomial iron after 12 weeks [28]. The mean increase was 0.7 g/dL and did not differ between CD and UC patients [28]. Nine subjects reported gastrointestinal adverse events (vomiting, nausea, constipation, tenesmus, diarrhea, epigastric pain, and intestinal bleeding), but all events were mild and none was clearly associated with the experimental treatment [28]. Almost all patients (29/30, 96.6%) completed the study, and 80% of participants took all the scheduled doses, suggesting a good tolerability of sucrosomial iron [28].

## 4. Discussion

Oral treatment of anemia is effective in IBD patients, and numerous efforts have been made to develop iron formulations aimed at reducing the number of adverse events, resulting in improved tolerance to oral administration. The main advantages of oral iron compared to intravenous iron are the reduced cost and the possibility of taking the drug at home, without the need to go to the hospital. The latter factor must be underlined and is even more important in the current historical period. Starting from December 2019, a new severe acute respiratory syndrome coronavirus 2 (SARS-CoV-2) has been identified in Wuhan, China, and in a few months, the disease associated with it (COVID-19) has globally spread [30]. As of 26 April 2020, 2,804,796 confirmed cases and 193,710 deaths for COVID-19 have been reported worldwide [31]. This disruptive health emergency has forced all hospitals to change their organization and reduce all non-essential activities, in order to prevent viral infection [32,33]. The COVID-19 outbreak has revolutionized the management of IBD patients, who require constant medical assistance and regular follow-ups. Most outpatient visits have been replaced by virtual monitoring, and many drug infusions have been rescheduled, including non-urgent intravenous iron administrations [32]. However, it is important to highlight that about 50% of IBD patients experience anemia and iron deficiency recurrences 10 and 11 months after intravenous treatment, respectively, thus suggesting the need for tight monitoring and adequate maintenance supplementation [34]. Lockdown has now been adopted in many countries, as a preventive strategy, and all individuals are recommended to stay home and to leave only in case of need [35]. We currently do not know the long-term effects of the pandemic on the management of IBD patients in terms of disease activity and re-exacerbation of symptoms and whether rescheduling of iron infusions can affect the outcomes of patients with anemia, but it is clear that a viable alternative for iron supplementation is available. Importantly, the oral treatment choice should be individualized, based on the risk–benefit ratio of each individual patient, and should be discussed with the patient. Several factors can influence uptake and absorption of oral iron (e.g., disease flares, ileocecal resections, and comorbidities (celiac disease and autoimmune gastritis)), and these factors should be considered, in order to improve iron therapy [13]. In fact, in post-surgical patients, there may be a concomitant deficiency of vitamin B12 or folate, which could contribute to the etiopathogenesis of anemia and should be supplemented [5]. Similarly, in IBD exacerbations, although iron stores are empty, ferritin levels may be normal due to the inflammatory stimulus; in these cases, a ferritin cutoff of 100 µg/L should be adopted, to administer iron supplementation [5]. Traditional ferrous oral iron is effective for the management of patients with martial deficiency anemia, but it is associated with a substantial increase in gastrointestinal adverse events, leading to poor adherence to therapy. Conversely, the new ferric iron complexes are effective for the treatment of anemic IBD patients and allow for less iron accumulation, minimizing adverse events and increasing drug tolerance. These data suggest that these new formulations may be superior to traditional ferrous iron, but in the absence of head-to-head trials directly comparing the drugs, it is not possible to draw definitive conclusions. Oral iron is a cost-effective option and could temporarily replace intravenous iron in patients with anemia during COVID-19 pandemic [13]. We hypothesize that IBD patients with mild (10 g/dL < Hb < 13 g/dL) and moderate (8 g/dL < Hb < 10 g/dL) anemia may first undergo oral iron supplementation, preferably with ferric maltol, as post-marketing studies confirmed the effective and safe drug profile (Figure 1) [25,26,27]. In contrast, in patients who are unresponsive to oral therapy, or in those with severe anemia (Hb < 8 g/dL), the intravenous formulation should be considered and not delayed.

## 5. Conclusions

Anemia is a frequent complication of patients with chronic IBD. Oral iron, including both the ferrous forms and the new ferric complexes, is effective and is recommended for the treatment of mild anemia. However, in the context of the COVID-19 health emergency, which requires maximum respect for the social-distancing rules and the reduction of nonessential hospital activities, the oral supplementation could be a suitable alternative for anemia therapy. We have learned an important lesson from the current pandemic and have provided an ambitious algorithm for the management of patients with martial-deficiency anemia, highlighting that patients can be managed at home and, only in urgent cases, in the hospital. Head-to-head comparative trials are needed to define which is the best oral iron formulation for the treatment of IBD patients with anemia.

## Figures and Tables

**Figure 1 jcm-09-01536-f001:**
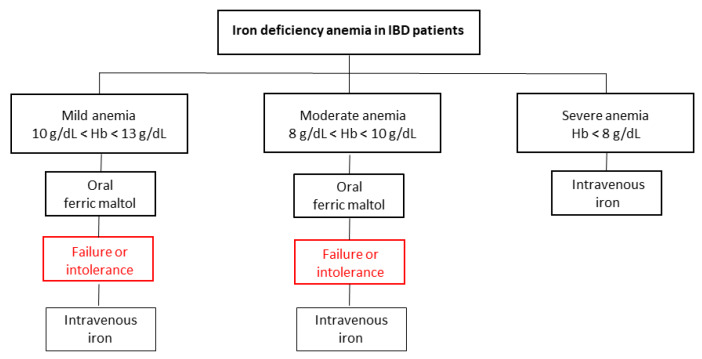
Proposed algorithm for the management of IBD patients with iron-deficiency anemia during COVID-19 pandemic.

**Table 1 jcm-09-01536-t001:** Main characteristics of ferrous oral iron studies for the treatment of anemia.

First Author	Study Design	Study Population	No. of Patients	Treatment	Results	Adverse Events
Lee [14]	Systematic review and meta-analysis	IBD	333	Oral ferrous iron IV iron	IV iron showed a higher improvement in Hb levels compared to oral iron	IV iron led to a lower rate of therapy discontinuation due to AEs compared to oral iron
Nielsen [15]	Systematic review	IBD	2906	Oral ferrous iron Oral ferric maltol IV iron	No difference between IV and oral iron in Hb increase in mild anemia	Milder AEs occurred more frequently in oral group than in IV group
Abhyankar [12]	Systematic review and meta-analysis	IBD	694	Oral ferrous iron IV iron	No difference between IV and oral iron in Hb response (Hb rise > 2 g/dL)	IV iron led to a lower rate of therapy discontinuation due to AEs compared to oral iron
Bonovas [4]	Systematic review and meta-analysis	IBD	694	Oral ferrous iron IV iron	IV iron showed a higher Hb response (Hb rise > 2 g/dL) compared to oral iron	GI AEs occurred more frequently in oral group than in IV arm
Reinisch [16]	Randomized open-label trial	IBD	338	Oral ferrous iron IV iron	Non-inferiority in Hb change between IV and oral iron was not proven	No difference in term of AEs was found between the study groups
Tolkien [17]	Systematic review and meta-analysis	Adult subjects	6831	Oral ferrous iron IV iron Placebo	/	More GI AEs occurred in oral group than in IV and placebo arms
Lugg [19]	Cross-sectional study	IBD	87	Oral ferrous iron IV iron	Median Hb change after oral iron was 7 g/L in CD patients and 4 g/L in UC patients	AEs occurred in 51% of the patients treated with oral iron

IBD: inflammatory bowel disease; n: number; IV: intravenous; GI: gastrointestinal; AEs: adverse events; Hb: hemoglobin; RCT: randomized controlled trial; CD: Crohn′s disease; UC: ulcerative colitis; /: not reported.

**Table 2 jcm-09-01536-t002:** Main characteristics of ferric oral iron studies for the treatment of anemia.

First Author	Study Design	Study Population	No. of Patients	Treatment	Results	Adverse Events
Gasche [20]	RCT	IBD	128	Ferric maltol Placebo	A mean Hb increase of 2.25 g/dL was found in the ferric maltol group vs. placebo arm	AEs were comparable between ferric maltol and placebo groups (58% vs. 72%)
Schmidt [21]	Prospective cohort study	IBD	97	Ferric maltol	86% of patients achieved normal Hb values after 64 weeks	Drug-related AEs were detected in 24% of patients
Farrell [22]	Systematic review	IBD	/	Ferric maltol Placebo	/	Fewer AEs (51% vs. 71%) and serious AEs (8% vs. 13%) occurred with ferric maltol compared to placebo
Cummings [23]	Prospective cohort study	IBD	30	Ferric maltol	62% of patients achieved normalized Hb levels after 3 months of therapy	/
Oppong [24]	Prospective cohort study	IBD	28	Ferric maltol	Ferric maltol was well-tolerated in 14/21 patients (66%)	The most common AEs was abdominal pain, no serious AE occurred
Howaldt [25]	RCT	IBD	250	Ferric maltol IV iron	Ferric maltol was not inferior to IV iron to achieve normalization or increase in Hb values of ≥2 g/dL	/
Howaldt [26]	RCT	IBD	250	Ferric maltol IV iron	Improvements in SF-36 and MCS scores were numerically higher with ferric maltol than IV iron	/
Howaldt [27]	RCT	IBD	250	Ferric maltol IV iron	Total costs per patient were lower in ferric maltol group than in IV arm	/
Abbati [28]	Prospective cohort study	IBD	30	Sucrosomial iron	Hb increased in 86% of patients after 3 months	44 AEs were recorded, but no AE was certainly related to the drug

IBD: inflammatory bowel disease; n: number; IV: intravenous; AEs: adverse events; Hb: hemoglobin; RCT: randomized controlled trial; SF-36: short form 36; MCS: mental component summary; /: not reported.

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
