# Peer review of "Oral Iron for IBD Patients: Lessons Learned at Time of COVID-19 Pandemic"

_jcm, 2020, doi:10.3390/jcm9051536_

Round 1

Reviewer 1 Report

The concept of this study is very important and this study could provide useful information to physicians who are managing IBD during this pandemic. However, there are many things that need to be changed prior to publication. First, there are numerous grammatical errors throughout the paper--so many that I cannot list them all here but can give a few examples. This paper should be reviewed thoroughly from a grammar standpoint prior to re-submission.

For example:

Line 15 "among which non-urgent iron infusions" is grammatically incorrect and missing a verb--it could be rephrased as "among which includes non-urgent iron infusions" or simply "including non-urgent iron infusions."

Line 23 "Crohn" should be "Crohn's" and inflammatory bowel disease should be plural since you are referring to two different types--UC and Crohn's.

Line 24 should say IBD mainly "causes" not "cause."

Again--these are just examples. There are grammatical errors in almost every line and too many to list them all here.

Here are my suggestions regarding content:

Line 28-blood loss is also an important cause of anemia in IBD patients and should be listed here as well

Line 30-should read "and is associated with an increased risk of hospitalization and surgery"--the studies quoted are association studies and do not prove that anemia causes these things

Line 53--this entire section needs a lot of work. There is essentially a listing of multiple studies but the differences between the studies and ways in which they are related are not relayed to the reader which makes the data presented difficult to interpret. A review should summarize and interpret the data and while this section summarizes, the interpretation part is minimal. Why did some of the studies come to different conclusions? This is important. Not all studies are created equal.

Line 66-I think its important to point out that ferritin level was increased in this meta-analysis despite the fact that hgb wasn't increased and to mention that this study is a meta-analysis (as opposed to the systematic review listed above)

Line 69-it is important to note that this study is included in the meta-analysis by Abhyankar and Moss as well as the systemic review by Nielsen--you list this as if this is a separate study but it is not. You are essentially accounting for it three times.

Line 76-again, this study is already included in the meta-analysis mentioned so you are essentially making it appear as if this is additional data but it is not. If you particularly want to highlight this study by mentioning it separately, you at least need to state that it is part of the meta-analysis.

Line 71-you failed to mention the highly important fact that this meta-analysis uses the exact same studies including in the meta-analysis by Abhyankar et al but they came to a different conclusion

Line 172-This is sentence is not very clear--why are celiac disease and autoimmune gastritis suddenly being mentioned? Are you concerned that IBD patients have these concomitant diseases which could affect iron absorption?

Line 176-You mention that there have been no head-to-head trials comparing ferric maltol to traditional iron supplements but then state in your algorithm that ferric maltol should be used preferentially. Why? I think the argument for this needs to be explained better if it is compelling enough to include in your algorithm.

Of note, I also noticed that the main meta-analysis quoted by the European Consensus in support of IV vs oral iron (Lee et al) was not included in your analysis. I would be interested to know why.

Author Response

Referee 1

The concept of this study is very important and this study could provide useful information to physicians who are managing IBD during this pandemic. However, there are many things that need to be changed prior to publication. First, there are numerous grammatical errors throughout the paper--so many that I cannot list them all here but can give a few examples. This paper should be reviewed thoroughly from a grammar standpoint prior to re-submission.

Reply: We thank the reviewer for the comment and his/her interest in this topic. We reviewed the grammar errors throughout the manuscript as requested.

For example:

Line 15 "among which non-urgent iron infusions" is grammatically incorrect and missing a verb--it could be rephrased as "among which includes non-urgent iron infusions" or simply "including non-urgent iron infusions."

Reply: Change made

Line 23 "Crohn" should be "Crohn's" and inflammatory bowel disease should be plural since you are referring to two different types--UC and Crohn's.

Reply: Change made

Line 24 should say IBD mainly "causes" not "cause."

Reply: Change made

Again--these are just examples. There are grammatical errors in almost every line and too many to list them all here.

Here are my suggestions regarding content:

Line 28-blood loss is also an important cause of anemia in IBD patients and should be listed here as well

Reply: Thank you for your comment. We have added “blood loss” as important cause of anemia in IBD patients.

Line 30-should read "and is associated with an increased risk of hospitalization and surgery"--the studies quoted are association studies and do not prove that anemia causes these things

Reply: Valid suggestion. We have made the appropriate change.

Line 53--this entire section needs a lot of work. There is essentially a listing of multiple studies but the differences between the studies and ways in which they are related are not relayed to the reader which makes the data presented difficult to interpret. A review should summarize and interpret the data and while this section summarizes, the interpretation part is minimal. Why did some of the studies come to different conclusions? This is important. Not all studies are created equal.

Reply: We thank the reviewer for this suggestion. We have substantially modified the paragraph, adding new data, eliminating overlaps, and specifying the characteristics of the different studies in order to allow a better understanding of the studies.

Line 66-I think its important to point out that ferritin level was increased in this meta-analysis despite the fact that hgb wasn't increased and to mention that this study is a meta-analysis (as opposed to the systematic review listed above)

Reply: We thank the reviewer for this comment. We have specified that the study by Abhyankar and Moss is a meta-analysis. We have also highlighted that ferritin levels were higher in patients treated with intravenous iron compared to those treated with oral iron.

Line 69-it is important to note that this study is included in the meta-analysis by Abhyankar and Moss as well as the systemic review by Nielsen--you list this as if this is a separate study but it is not. You are essentially accounting for it three times.

Reply: We thank the reviewer for his suggestion. We removed this study to avoid repeating the data.

Line 76-again, this study is already included in the meta-analysis mentioned so you are essentially making it appear as if this is additional data but it is not. If you particularly want to highlight this study by mentioning it separately, you at least need to state that it is part of the meta-analysis.

Reply: We appreciate the comment of the reviewer. We have specified that the work is part of the meta-analyses by Abhyankar and Moss and by Bonovas et al. as requested.

Line 71-you failed to mention the highly important fact that this meta-analysis uses the exact same studies including in the meta-analysis by Abhyankar et al but they came to a different conclusion

Reply: We thank the reviewer for the suggestion. He rephrased our sentence, clarifying that the meta-analysis by Bonovas and colleagues used the same studies included in the meta-analysis by Abhyankar and Moss.

Line 172-This is sentence is not very clear--why are celiac disease and autoimmune gastritis suddenly being mentioned? Are you concerned that IBD patients have these concomitant diseases which could affect iron absorption?

Reply: We have clarified our sentence as requested: “Several factors can influence uptake and absorption of oral iron (e.g. disease flares, ileo-cecal resections, and comorbidities (celiac disease and autoimmune gastritis)), and should be considered to improve iron therapy”

Line 176-You mention that there have been no head-to-head trials comparing ferric maltol to traditional iron supplements but then state in your algorithm that ferric maltol should be used preferentially. Why? I think the argument for this needs to be explained better if it is compelling enough to include in your algorithm.

Reply: We thank the reviewer for the comment. We have further explained the rationale for the use of ferric maltol compared to traditional iron. The main reason for this decision is due to its mechanism of action, which should guarantee a better safety profile and a theoretical advantage in terms of adverse events.

Of note, I also noticed that the main meta-analysis quoted by the European Consensus in support of IV vs oral iron (Lee et al) was not included in your analysis. I would be interested to know why.

Reply: We thank the reviewer for his/her comment. We added the meta-analysis by Lee et al. in our review: “A meta-analysis by Lee et al. analyzed data from 333 IBD patients recruited in 3 randomized clinical trials, assessing efficacy and safety of oral and intravenous iron formulations [14]. The primary outcome was the mean difference in Hb at the end of study compared to baseline. A greater improvement in Hb levels was achieved after infusion therapy compared to oral administration with a mean difference of 6.8 g / L (CI 0.9, 12.7, p = 0.02) [14]. A higher increase in serum ferritin levels was also found in the intravenous group compared to the oral one with a mean difference of 109.7 μg / L (CI 5.37, 214, p = 0.04). Adverse events leading to therapy discontinuation mostly occurred in patients treated with oral iron (odds ratio 6.2, CI 2.21, 17.1), while no difference between the two groups was detected in terms of quality of life [14].”

Reviewer 2 Report

This is a well written narrative review article with a management algorithm.. 
it is not clear why the authors didn't go ahead a do a full systematic review with PRISMA guidelines? Particularly if advocating a management algorithm for the covid-era 

Having said that, the review is written well and the evidence and papers used are presented and it benefits from a distinguished authorship who's opinion on the topic matter are likely to be considered and cited. 

Author Response

Referee 2

This is a well written narrative review article with a management algorithm..

it is not clear why the authors didn't go ahead a do a full systematic review with PRISMA guidelines? Particularly if advocating a management algorithm for the covid-era

Reply: We gratefully thank the reviewer for her/his positive comment. We performed a narrative review to provide an overview of the current literature, as requested by the editor because this is a special issue.

Having said that, the review is written well and the evidence and papers used are presented and it benefits from a distinguished authorship who's opinion on the topic matter are likely to be considered and cited.

Reply: We thank the reviewer for this comment.

Reviewer 3 Report

This manuscript presents important information for the management of patients with IBD under pandemic conditions. However, it is necessary to add a few arguments.

In patients with inflammatory bowel disease, especially Crohn's disease, they often experience surgical treatment during its course. In particular, patients who have undergone a ileo-cecal resection may develop macrocytic anemia associated with vitamin B12 deficiency. In other words, anemia occurring in inflammatory bowel disease is influenced by a combination of factors other than iron deficiency anemia associated with bleeding. Although the causes of anemia in this review are limited to iron deficiency, this complex of factors should be discussed for practical clinical application.

In addition, in patients with persistent chronic inflammation, high ferritin is experienced even when serum iron is low. The authors should also discuss the need for iron administration in such cases.

Author Response

Referee 3

This manuscript presents important information for the management of patients with IBD under pandemic conditions. However, it is necessary to add a few arguments.

In patients with inflammatory bowel disease, especially Crohn's disease, they often experience surgical treatment during its course. In particular, patients who have undergone a ileo-cecal resection may develop macrocytic anemia associated with vitamin B12 deficiency. In other words, anemia occurring in inflammatory bowel disease is influenced by a combination of factors other than iron deficiency anemia associated with bleeding. Although the causes of anemia in this review are limited to iron deficiency, this complex of factors should be discussed for practical clinical application.

Reply: Thank you for your comment. We have added “vitamin B12 deficiency” as important cause of anemia in IBD patients and we have discussed the therapeutic approach to patients with multifactorial anemia, including vitamin deficiencies and disease exacerbations: “In fact, in operated patients there may be a concomitant deficiency of vitamin B12 or folate which could contribute to the etiopathogenesis of anemia and should be supplemented [5]. Similarly, in IBD exacerbations, although iron stores are empty, ferritin levels may be normal due to the inflammatory stimulus; in these cases a ferritin cut-off of 100  µg/L should be adopted to administer iron supplementation [5]”.

In addition, in patients with persistent chronic inflammation, high ferritin is experienced even when serum iron is low. The authors should also discuss the need for iron administration in such cases.

Reply: As specified in the response to the previous comment, we discussed the need for iron administration in patients who do not have low ferritin values.

Round 2

Reviewer 1 Report

I commend the authors on substantial improvements to the manuscript, both in terms of grammar as well as content.

There are a few minor suggestions that I have:

Line 53-given the large amount of data presented in this paragraph, this paragraph would be improved with a sentence at the end summarizing what the data overall suggests in terms of oral vs IV efficacy and tolerance 

Line 59-should say "the treatment of anemia" rather than "the anemia treatment"

Line 70-subject is missing for the verb "allowed." Should be "allowed for the attainment of" or "allowed for patients to obtain"

Line 84: I would recommend "included by colleagues and concluded that intravenous iron" rather than use of the colon. 

Line 94: I would clarify that the point of the meta-analysis was to evaluate for side effects of iron therapy since this is different from the studies above (which also looked for efficacy)

Line 185 "discussed" would be a more appropriate word than "shared"

Lines 29 and 187 would recommend "post-surgical" rather than "operated" patients

Author Response

Referee 1

I commend the authors on substantial improvements to the manuscript, both in terms of grammar as well as content.

Reply: We gratefully thank the reviewer for the comment and the positive feedback.

There are a few minor suggestions that I have:

Line 53-given the large amount of data presented in this paragraph, this paragraph would be improved with a sentence at the end summarizing what the data overall suggests in terms of oral vs IV efficacy and tolerance 

Reply: We thank the reviewer for the comment. We have added the following sentence at the end of this section as requested: “Overall, both oral and intravenous iron are effective for the treatment of martial deficiency anemia, but the greater percentage of adverse events in orally treated patients invalidates patient tolerance and adherence to therapy.”.

Line 59-should say "the treatment of anemia" rather than "the anemia treatment"

Reply: Change made.

Line 70-subject is missing for the verb "allowed." Should be "allowed for the attainment of" or "allowed for patients to obtain"

Reply: We thank the reviewer for the comment. We have made the appropriate change.

Line 84: I would recommend "included by colleagues and concluded that intravenous iron" rather than use of the colon. 

Reply: We have modified our sentence as recommended.

Line 94: I would clarify that the point of the meta-analysis was to evaluate for side effects of iron therapy since this is different from the studies above (which also looked for efficacy)

Reply: We thank the reviewer for the comment. We have clarified that the meta-analysis evaluated the safety profile of oral and intravenous iron as requested.

Line 185 "discussed" would be a more appropriate word than "shared"

Reply: Change made

Lines 29 and 187 would recommend "post-surgical" rather than "operated" patients

Reply: Change made.